# Anticariogenic Activity of Celastrol and Its Enhancement of Streptococcal Antagonism in Multispecies Biofilm

**DOI:** 10.3390/antibiotics12081245

**Published:** 2023-07-28

**Authors:** Hao Li, Chenguang Niu, Junyuan Luo, Zhengwei Huang, Wei Zhou

**Affiliations:** 1Department of Endodontics, Shanghai Ninth People’s Hospital, Shanghai Jiao Tong University School of Medicine, College of Stomatology, Shanghai Jiao Tong University, 500 Quxi Road, Shanghai 200011, China; lihao106@shsmu.edu.cn (H.L.);; 2National Center for Stomatology, National Clinical Research Center for Oral Diseases, Shanghai Key Laboratory of Stomatology, No. 639, Zhizaoju Road, Shanghai 200011, China

**Keywords:** dental caries, celastrol, *Streptococcus mutans*, *Streptococcus sanguinis*, *Streptococcus gordonii*, oral streptococcal antagonism

## Abstract

Dental caries is a chronic disease resulting from dysbiosis in the oral microbiome. Antagonism of commensal *Streptococcus sanguinis* and *Streptococcus gordonii* against cariogenic *Streptococcus mutans* is pivotal to keep the microecological balance. However, concerns are growing on antimicrobial agents in anticaries therapy, for broad spectrum antimicrobials may have a profound impact on the oral microbial community, especially on commensals. Here, we report celastrol, extracted from Traditional Chinese Medicine’s *Tripterygium wilfordii* (TW) plant, as a promising anticaries candidate. Our results revealed that celastrol showed antibacterial and antibiofilm activity against cariogenic bacteria *S. mutans* while exhibiting low cytotoxicity. By using a multispecies biofilm formed by *S. mutans* UA159, *S. sanguinis* SK36, and *S. gordonii* DL1, we observed that even at relatively low concentrations, celastrol reduced *S. mutans* proportion and thereby inhibited lactic acid production as well as water-insoluble glucan formation. We found that celastrol thwarted *S. mutans* outgrowth through the activation of pyruvate oxidase (SpxB) and H_2_O_2_-dependent antagonism between commensal oral streptococci and *S. mutans*. Our data reveal new anticaries properties of celastrol that enhance oral streptococcal antagonism, which thwarts *S. mutans* outgrowth, indicating its potential to maintain oral microbial balance for prospective anticaries therapy.

## 1. Introduction

More than half a century has passed since the widespread applications of fluoride and sealant therapy. Yet, dental caries remains one of the most prevalent chronic oral diseases [1,2]. According to the global burden of disease study released by Lancet in 2017, the prevalence of permanent dental caries ranked first among 328 diseases, posing a major health burden for individuals throughout their lifetime. Therefore, there is an urgent need for preventative and curative measures of anticaries therapy.

Based on modern etiologic studies, dental caries is a chronic disease resulting from a dysbiosis in the oral microbiome [3,4,5]. Interspecies interactions among bacterial species, including synergistic interactions and antagonistic interactions, are pivotal to keep the microecological balance between commensals and pathogens, which contributes to oral health [6]. The antagonism among the microecology of oral streptococci is a representative example associated with dental caries. *Streptococcus sanguinis* and *Streptococcus gordonii*, which belong to Mitis group streptococci, are the most ubiquitous and abundant commensal colonizers of tooth surfaces [7,8]. In their role as primary colonizers, *S. sanguinis* and *S. gordonii* can compete against cariogenic species *Streptococcus mutans* by secreting hydrogen peroxide (H_2_O_2_) via pyruvate oxidase (SpxB) [9,10]. Such antagonism between commensals and *S. mutans* is beneficial for preventing the overgrowth of cariogenic species [11]. However, ecological pressures such as sugar intake, poor oral hygiene, or reduced salivary flow can contribute to the reduced proportions of the commensal species and result in the selection of cariogenic species such as *S. mutans* [12,13]. The overgrowth of *S. mutans* contributes to the occurrence of caries by producing lactic acid [14,15], as well as water-insoluble glucan, the main component of exopolysaccharide (EPS), which facilitates bacterial adhesion [16,17,18]. In this context, managing the antagonism among the microecology of oral streptococci that boost commensals and keep pathogenic species in check could be an ideal approach to maintain the symbiosis of oral microecology for prospective anticaries therapy [19,20].

Due to the bacterial etiology of dental caries, the application of antibacterial agents has long been considered an effective strategy for anticaries therapy [21]. Antimicrobial agents with a broad spectrum of activity such as chlorhexidine, quaternary ammonium salts, and antimicrobial peptides are employed in anticaries therapies [21,22,23]. The use of these antibacterial agents aims at decreasing the total bacterial load. However, the impact of current antibacterial agents on interspecies interaction and oral microecology remains to be elucidated. In fact, concerns are growing on antibacterial agents in anticaries therapy, for broad spectrum antimicrobials may have a profound impact on the oral microbial community, especially on commensals [24,25]. Moreover, recent evidence has shown that chlorhexidine treatment induced profound shifts in the bacterial composition of oral biofilm and led to a higher abundance of pathobiont strains, which included *Streptococcus* species [26]. Although the precise mechanism remains incompletely understood, such a composition shift caused by chlorhexidine may result in limited clinical efficacy for anticaries therapy. Therefore, there is an urgent need to unravel the impact of antimicrobials on oral microbiomes and microbial interactions.

The identification of bioactive compounds derived from herbal products has attracted considerable interest in recent years for their therapeutic potential. One representative example is celastrol, extracted from Traditional Chinese Medicine’s *Tripterygium wilfordii* (TW) plant, which is recognized as a star bioactive molecule for its potential in treating multiple chronic diseases [27,28]. Celastrol (3-hydroxy-9β,13α-dimethyl-2-oxo-24,25,26-trinoroleana-1(10),3,5,7-tetraen-29-oic acid) is a pentacyclic triterpenoid (Figure 1), and it possesses a variety of biological activities, including anti-inflammatory [29], anticancer [30], and anti-obesity [31]. In recent years, the antibacterial potential of celastrol against common pathogens (e.g., *Staphylococcus aureus* and *Klebsiella pneumoniae*) has also been discovered [32,33,34]. Recent investigation has shown that celastrol as a promising antibacterial candidate exhibits a broad spectrum of activity by acting on multiple targets, including damaging the cytoplasmic membrane, disrupting cell respiration, and affecting the transport of solutes [32].

Despite the growing interest in celastrol as an antibacterial candidate, no previous study has characterized the anticaries potential of celastrol. Meanwhile, in the case of dental caries, in addition to antibacterial properties, the impacts on oral microbial interactions and oral microecological balance are also important considerations for new anticaries agents [35]. Therefore, this study aimed to explore the anticaries potential of celastrol, including its antibacterial activity and its impact on oral microbial interactions. The cytotoxicity and the antibacterial activity of celastrol against oral streptococci were evaluated. By using a multispecies biofilm formed by well-characterized oral streptococci including *S. mutans* UA159 (UA159), *S. sanguinis* SK36 (SK36), and *S. gordonii* DL1 (DL1), we investigated the impact of celastrol on microbial interactions in oral streptococci. Specifically, we found that celastrol at high concentrations (2, 4, and 8 μg/mL) exhibited antibacterial activity against oral streptococci, including cariogenic *S. mutans*. Meanwhile, when at relatively low concentrations, celastrol reduced the proportion of *S. mutans* by promoting streptococcal antagonism and inhibited lactic acid production as well as water-insoluble glucan formation in a multispecies biofilm.

## 2. Results

### 2.1. The Cytotoxicity and Antibacterial Properties of Celastrol against Oral Streptococci

To evaluate the antibacterial activity of celastrol against oral streptococci, the minimum inhibitory concentration (MIC) and minimum bactericidal concentration (MBC) were evaluated. The MIC was 4 μg/mL for all the test strains including *S. mutans* UA159 (UA159), *S. sanguinis* SK36 (SK36), and *S. gordonii* DL1 (DL1) (Figure 2a). The MBC was 8 μg/mL for SK36 and DL1 and 16 μg/mL for UA159 (Figure 2a). Then, the biosafety of celastrol was determined by the cytotoxicity of DMAEM upon human oral keratinocyte (HOK) cells. There was no significant difference in cell activity after exposure with 0.25–8 μg/mL of celastrol for 24 h, while celastrol induced a significant reduction in the cell viability at 16–256 μg/mL, when compared to the control group (0 μg/mL) (Figure 2b). Considering the cytotoxicity, concentrations under 8 μg/mL were selected for the subsequent research.

For planktonic cells, celastrol at 2, 4, and 8 μg/mL showed an inhibitory effect on the growth of UA159, SK36, and DL1, while at relatively low concentrations (0.25, 0.5, and 1 μg/mL), the inhibitory effect of celastrol on growth vanished (Figure 2c–e). Quantitative biofilm analyses were performed by colony counting after celastrol exposure. In the UA159 monospecies biofilm, as well as in multispecies biofilm, celastrol at 2, 4, and 8 μg/mL significantly reduced the number of living bacteria in a dose-dependent manner. In 24 h biofilm, celastrol at 2, 4, and 8 μg/mL resulted in a 3.3–4.8 log_10_ reduction against the UA159 biofilm and a 3.6–5.2 log_10_ reduction against multispecies biofilm (Figure 2f). In biofilm cultured for 48 h, celastrol at 2, 4, and 8 μg/mL resulted in a 4.1–5.3 log_10_ reduction against UA159 biofilm and a 3.1–5.7 log_10_ reduction against multispecies biofilm (Figure 2g). For biofilm cultured for 72 h, celastrol at 2, 4, and 8 μg/mL led to a 2.2–5.7 log_10_ reduction against UA159 biofilm and a 3.9–5.6 log_10_ reduction against multispecies biofilm (Figure 2h). For celastrol at low concentrations, we only detected a 2.5 log_10_ reduction against UA159 biofilm and a 1.4 log_10_ reduction against multispecies biofilm treated with 1 μg/mL celastrol for 24 h (Figure 2f). While celastrol at 0.25 and 0.5 μg/mL showed no effect on the number of viable bacteria, in mature biofilm cultured for 48 h and 72 h, we noticed that celastrol at low concentrations (0.25, 0.5 and 1 μg/mL) showed no significant effect on bacterial viability in both the UA159 monospecies biofilm and multispecies biofilm groups (Figure 2g,h).

Taken together, our results revealed that celastrol at 2, 4, and 8 μg/mL exhibited potent antibacterial activity against oral streptococci, while at low concentrations (0.25, 0.5, and 1 μg/mL) it showed no obvious antibacterial ability.

### 2.2. Celastrol, at Relatively Low Concentrations, Inhibited Lactic Acid Production and Water-Insoluble Glucan Formation in Multispecies Biofilm

In the UA159 biofilm, celastrol treatments at 2, 4, and 8 μg/mL significantly reduced lactic acid production. Celastrol at 2, 4, and 8 μg/mL reduced lactic acid production by more than 90% compared to the control in 24, 48, and 72 h biofilm (Figure 3a–c). While at low concentrations (0.25, 0.5, and 1 μg/mL), the inhibitory effects of celastrol on acid production were impaired (Figure 3a–c), especially in mature biofilm cultured for 72 h, celastrol at low concentrations showed no obvious inhibitory ability on acid production (Figure 3c). Next, quantification of water-insoluble glucan was performed. In the UA159 biofilm cultured for 24, 48, and 72 h, celastrol at high concentrations (2, 4, and 8 μg/mL) significantly inhibited the water-insoluble glucan production by more than 50% (Figure 3d–f). However, low concentrations (0.25, 0.5, and 1 μg/mL) showed no obvious inhibitory ability on water-insoluble glucan production in the UA159 biofilm (Figure 3d–f).

In multispecies biofilm, we detected that celastrol inhibited acid production and water-insoluble glucan production in the range (concentrations) from 0.25 to 8 µg/mL. Celastrol at high concentrations (2, 4, and 8 μg/mL) inhibited acid production in 24, 48, and 72 h biofilm by more than 90% (Figure 3a–c). Meanwhile, when at low concentrations (0.25, 0.5, and 1 μg/mL), celastrol treatment also reduced acid production by 18.7–81.3% in 24, 48, and 72 h biofilm (Figure 3a–c). For water-insoluble glucan production, celastrol at high concentrations (2, 4, and 8 μg/mL) inhibited water-insoluble glucan production in 24, 48, and 72 h biofilm by more than 90% (Figure 3d–f). When at relatively low concentrations (0.25, 0.5, and 1 μg/mL), celastrol could still lead to a 11–32% reduction in water-insoluble glucan production against 24, 48, and 72 h multispecies biofilm (Figure 3d–f).

It is worth mentioning that when comparing the multispecies biofilm and UA159 biofilm without celastrol treatment (0 μg/mL), there was no statistical difference in lactic acid production (Figure 3a–c) and water-insoluble glucan production (Figure 3d–f) capacity between the two groups. Celastrol at 2, 4, and 8 μg/mL simultaneously reduced the acid and water-insoluble glucan production in both the UA159 biofilm and multispecies biofilm, consistent with the potent antibacterial activity of celastrol at high concentrations. However, at relatively low concentrations (0.25, 0.5, and 1 μg/mL), celastrol only exhibited inhibitory effects on multispecies biofilm, as the lactic acid (Figure 3a–c) and water-insoluble glucan production (Figure 3d–f) after treatment were significantly lower when compared to UA159 biofilm groups. Specifically, compared with the UA159 biofilm, after 24 h of treatment, celastrol at 0.25, 0.5, and 1 μg/mL led to a 3–27% reduction in acid production (Figure 3a) and a 16–23% reduction in water-insoluble glucan production (Figure 3d) in multispecies biofilm. For 48 h of treatment, celastrol at 0.25, 0.5, and 1 μg/mL led to a 34–40% reduction in acid production (Figure 3b) and a 16–20% reduction in water-insoluble glucan production (Figure 3e) in multispecies biofilm. In addition, after 72 h of treatment, celastrol at 0.25, 0.5, and 1 μg/mL led to a 35–60% reduction in acid production (Figure 3c) and a 14–20% reduction in water-insoluble glucan production (Figure 3f) in multispecies biofilm, when compared with the UA159 biofilm.

Scanning electron microscopy (SEM) images further confirmed the difference in the inhibitory ability of celastrol on multispecies biofilm and UA159 biofilm (Figure 3g). A rough biofilm structure formed by mature microbial aggregate containing a dense extracellular matrix and aggregated bacteria (red arrows) could be observed in the two control groups without celastrol treatment (0 μg/mL). After exposure to 2 μg/mL celastrol, visually reduced bacteria could be observed in both multispecies biofilm and UA159 biofilm, consistent with the reduction in viable cell count shown in Figure 2f. We also observed a dramatic reduction in extracellular matrix, which may be due to the inhibitory effects of celastrol on water-insoluble glucan production, as well as EPS collapse in loose and thin biofilm during SEM sample preparation [36]. Bacterial lysate fragments (yellow arrows) could be observed in multispecies biofilm and UA159 biofilm after exposure to 2 μg/mL celastrol, suggesting celastrol at this concentration had a strong antibacterial effect which led to bacterial lysis. In low-concentration groups (0.25, 0.5, and 1 μg/mL), no apparent morphological changes were observed in the UA159 biofilm compared to the untreated control. However, in multispecies biofilm, celastrol significantly inhibited water-insoluble glucan formation without affecting bacterial growth, resulting in loose and thin biofilm structures (Figure 3g).

### 2.3. Celastrol Reduced S. mutans Proportion in Multispecies Biofilm by Promoting Streptococcal Antagonism

*S. mutans* is the major species producing lactic acid and water-insoluble glucan in oral streptococci [12,37]. Therefore, we speculated that the reduction in lactic acid and water-insoluble glucan production in multispecies biofilm with celastrol treatment was due to the reduction in *S. mutans*. To test this, we detected the bacterial composition of biofilm treated with celastrol for 24, 48, and 72 h. In the control group without celastrol treatment (0 μg/mL), there was an increasing proportion of UA159, and the percentage reached from 37.54% at 24 h to 50.02% at 72 h, while both SK36 and DL1 proportions decreased over time. However, treatment of celastrol at 0.25, 0.5, and 1 μg/mL led to a 3- to 12-fold reduction in the proportion of UA159 in the 24 h biofilm. Treatments of mature biofilm cultured for 48 h and 72 h with the same concentrations of celastrol also resulted in a 3- to 16-fold reduction in the proportion of UA159 (Figure 4a and Appendix A). As for commensal Streptococci, treatment of celastrol at 0.25, 0.5, and 1 μg/mL led to a 1.9- to 3.5-fold increase in the proportion of SK36 for 24, 48, and 72 h biofilm (Figure 4a and Appendix A). In low-concentration groups, the relative abundance of SK36 in 24, 48, and 72 h biofilm exceeded 60%. Treatment of celastrol at low concentrations led to a six- to three-fold reduction in the proportion of DL1 in 24 h and 48 h biofilm. In 72 h mature biofilm, the DL1 proportion remained almost unchanged (about 23% in abundance) (Figure 4a and Appendix A).

These results revealed that celastrol at low concentrations reduced *S. mutans* proportion and boosted commensal *S. sanguinis* in multispecies biofilm, leading to the reduction in lactic acid and water-insoluble glucan. This phenomenon cannot be simply explained by the antibacterial activity of celastrol, since celastrol at these concentrations (0.25, 0.5, and 1 μg/mL) showed no obvious inhibitory effect on bacterial growth (Figure 2c–e). Therefore, we speculated that celastrol may regulate microbial composition by enhancing streptococcal antagonism. We analyzed the antagonism of SK36 and DL1 against UA159 under different concentrations of celastrol. In the control group (0 μg/mL), SK36 and DL1 showed slight inhibitory effects against UA159, since the lawn of *S. mutans* became thinner at the intersection without an obvious proximal zone of inhibition (Figure 4b). At low celastrol concentrations (0.5 and 1 μg/mL), both commensal strains inhibited the growth of UA159, with SK36 showing a stronger inhibitory effect on UA159 for the larger area of inhibition region (Figure 4c,d). However, at high celastrol concentration (2 μg/mL), SK36 and DL1 no longer inhibited UA159 for limited growth (Figure 4e). These results confirmed that celastrol could reduce *S. mutans* proportion in multispecies biofilm by promoting streptococcal antagonism, rather than killing UA159 directly.

### 2.4. Celastrol Thwarted S. mutans Outgrowth by Promoting H_2_O_2_-Dependent Antagonism Mediated by SpxB

H_2_O_2_ production mediated by SpxB in commensal streptococci is well known to inhibit the growth of *S. mutans*. To examine whether the growth inhibition of UA159 after celastrol treatment was mediated by H_2_O_2_-dependent antagonism, we tested the H_2_O_2_ production following celastrol treatment. We found that celastrol treatments (0.25, 0.5, and 1 μg/mL) significantly increased H_2_O_2_ production in 24 h and 48 h multispecies biofilm (Figure 5a). Consistent with the increased level of H_2_O_2_ production, the expression profile of *spxB* also showed increase expression in both SK36 (Figure 5b) and DL1 (Figure 5c) following celastrol treatments. In both *S. sanguinis* and *S. gordonii*, SpxB catalyzes the oxidative decarboxylation of pyruvate to H_2_O_2_ and the high energy metabolite acetyl-phosphate [38]. Therefore, pyruvate depletion reflects the activity of SpxB [7,38]. In SK36 and DL1, we noted significant depletion of pyruvate in response to celastrol treatments (Figure 5d,e), consistent with increased H_2_O_2_ production and *spxB* expression.

We further explored whether the inhibition of celastrol against UA159 can be compromised by the addition of antioxidant catalase to multispecies biofilm. As shown in Appendix A, supplementation with catalase at 10 μg/mL effectively reduced H_2_O_2_ in biofilm treated with celastrol (Appendix A), without affecting the total bacteria count (Appendix A). Then, bacterial composition was determined. In biofilm without celastrol treatments, there was no statistical difference in the proportion of UA159 in the presence or absence of catalase (Figure 6a and Appendix A). In celastrol-treated groups, however, the proportions of UA159 in the multispecies biofilm were significantly increased by adding catalase (Figure 6a and Appendix A). In the 0.25, 0.5, and 1 μg/mL celastrol groups, the proportion of UA159 increased from 2% to 18%, 9% to 20%, and 0.3% to 7.5%, respectively. Consistent with the increased proportion of UA159, we noted that in multispecies biofilm treated with celastrol at low concentrations (0.25, 0.5, and 1 μg/mL), the supplementation of catalase led to a 36–42% increase in lactic acid production (Figure 6b), as well as a 28–45% increase in water-insoluble glucan production (Figure 6c).

The enhancement of streptococcal antagonism mediated by celastrol was further confirmed by SK36 Δ*spxB* and DL1 Δ*spxB* mutants. Due to the defects in H_2_O_2_ generation, SK36 Δ*spxB* and DL1 Δ*spxB* failed to inhibit UA159 in the presence or absence of celastrol (Appendix A). We next evaluated lactic acid production and water-insoluble glucan production in multispecies biofilm cultured with SK36 Δ*spxB* and DL1 Δ*spxB*. When compared to biofilm cultured with SK36 and DL1, no significant differences in acid and water-insoluble glucan production were observed between the two control groups (0 μg/mL), while there was a 24–48% increase in lactic acid production (Figure 6d) and a 26–38% increase in water-insoluble glucan production (Figure 6e) after treatments with low concentrations of celastrol (0.25, 0.5, and 1 μg/mL) in multispecies biofilm cultured with SK36 Δ*spxB* and DL1 Δ*spxB.*

## 3. Discussion

Contemporary paradigms of caries etiology focus on the microecology of the oral biofilm; therefore, there is currently a major emphasis on opportunities and challenges associated with anticaries approaches to shift the dental biofilm from a state of dysbiosis to a state of symbiosis [39,40]. Here, we reported that celastrol, as a bioactive component of Traditional Chinese Medicine, could be a promising therapeutic candidate for anticaries strategy.

In this study, we used a well-characterized multispecies ecological biofilm model to evaluate the anticaries performance of novel antibacterial agents [41,42,43], which were composed of oral streptococci closely related to the occurrence of caries. Among them, *S. mutans* UA159 is the most important cariogenic bacterium, while *S. sanguinis* SK36 and *S. gordonii* DL1 are generally recognized as benign commensals in caries development [44,45]. The antibacterial profile revealed that celastrol at 2, 4, and 8 μg/mL exhibited potent antibacterial and antibiofilm activity against both UA159 and multispecies biofilms, which is consistent with recent findings suggesting its potential in tackling infectious disease by exerting bactericidal activity at certain concentrations [32,33,34].

As shown in Figure 3g, bacterial lysate fragments could be observed in biofilm treated with 2 μg/mL celastrol. We speculate that this phenomenon may be due to an action of celastrol on the integrity of the cell membrane, which leads to permeability changes and the release of the cytoplasmic content. Similar results have been observed in treatments of other triterpenoid compounds including zeylasterone, oleanolic acid, and ursolic acid, for which an action on the bacterial cell membrane has also been suggested [46,47], although further investigations are required to elucidate the precise mechanism of action for celastrol.

Concerns are growing on the impact of antimicrobials on oral microbial communities. Antimicrobials with a broad spectrum of activities already have been already shown to cause dysbiosis by altering the microbial interactions between pathogens and commensals [25,48]. Moreover, in reality, due to the flow of saliva and food intake, the actual concentration of anticaries agents in the oral environment is difficult to maintain at the concentrations with biological activity [49,50]. Therefore, it is important to start considering the disruption of microbial interactions mediated by antibacterial agents, especially at concentrations below MIC [51]. In this study, we revealed the favorable modification of the microbial composition mediated by celastrol at concentrations below MIC. Celastrol at relatively low concentrations (0.25, 0.5, and 1 μg/mL) reduced the *S. mutans* proportion by activating a key modulator involved in H_2_O_2_-dependent antagonism, SpxB, and led to reduced acid production and water-insoluble glucan production. These results highlighted the anticaries potential of celastrol as an “ecological regulator” by promoting Streptococcal antagonism. Moreover, our results showed that the inhibitory effect of celastrol on lactic acid and water-insoluble glucan production was significantly impaired by the addition of catalase, suggesting that H_2_O_2_ secreted by commensals can greatly influence the outcome of celastrol treatment.

The mechanisms of SpxB activation mediated by celastrol remain to be unraveled. In common colonizer *Streptococcus pneumoniae*, SpxB prevents ATP depletion during oxidative stress conditions, by allowing the conversion of acetyl phosphate into ATP [52,53,54]. Therefore, SpxB activation is required for the growth of *spxB*-encoding streptococci under oxidative stress. According to recent studies, the structure of celastrol holds a highly redox-active paraquinone methide moiety, which leads to oxidative stress by forming reactive oxygen species (ROS) in eukaryotic cells [55,56,57]. Therefore, the upregulation of oxidative stress may also exist in oral streptococci treated with celastrol. According to the results in Appendix A, the addition of antioxidant catalase did not affect the number of viable bacteria in biofilm treated with celastrol, indicating that low concentrations of celastrol induced non-lethal oxidative stress. Therefore, we speculate that celastrol activates SpxB by generating non-lethal oxidative stress, although this issue requires further investigation.

According to the results of the competition assay shown in Figure 4b–d, celastrol activated the antagonism of DL1 and SK36 against UA159, respectively, while in multispecies biofilm, SK36 outcompeted both UA159 and DL1, becoming the dominant species (Figure 4a). As reported in previous research, in caries lesions, *S. gordonii* diminished in plaque biofilm, while *S. mutans* and *S. sanguinis* persisted [58]. Another study showed that *S. mutans* always out-competes *S. gordonii* on teeth in vivo, independent of diet, sequence of inoculation, and strains [59]. Based on these findings, we speculate that the reason why SK36 achieved growth advantage when treated with celastrol is that SK36 possessed a stronger antagonistic effect against UA159 than DL1.

The limitations of this study should be addressed in future investigations. As a bioactive compound with multiple pharmacological activities, the limitation of celastrol related to stability and biotoxicity has been described [60,61]; therefore, it is necessary to verify the biosafety and stability of celastrol using in vivo models. Our results revealed that, in addition to the antibacterial effect, the anticaries capacity of celastrol also owes to the enhancement in streptococcal antagonism by activating SpxB. Yet, the specific mechanism of *spxB* activation by low concentrations of celastrol still needs further experimental investigation. Moreover, our data only preliminarily indicate the potential enhancement of antagonism on selected oral streptococci in batch culture. Due to the complex environment and microbial composition of dental biofilms in reality, the exact efficacy of celastrol should be further determined in vivo with an optimal delivery method (such as rinse, dentifrice, or varnish, etc.), and further work is needed to focus on the characteristics of species interactions, as well as microbial composition alterations under celastrol treatment to validate its clinical effectiveness for caries treatment and prevention.

## 4. Conclusions

In this study, we evaluated the anticaries potential of celastrol. Our findings revealed that celastrol exhibited low cytotoxicity. At high concentrations (2, 4, and 8 μg/mL), celastrol showed potent antibacterial activity against oral streptococci, including cariogenic *S. mutans*, while no obvious antibacterial activity was observed at relatively low concentrations (0.25, 0.5, and 1 μg/mL). In multispecies biofilm, celastrol at relatively low concentrations (0.25, 0.5, and 1 μg/mL) reduced *S. mutans* proportion and thereby inhibited the acidogenicity as well as water-insoluble glucan formation. The results obtained in this study demonstrated that celastrol activated pyruvate oxidase (SpxB) and promoted the H_2_O_2_-dependent antagonism of *S. sanguinis* and *S. gordonii* against *S. mutans*. Our data reveal a new biological property of celastrol that enhances oral streptococcal antagonism thwarting *S. mutans* outgrowth, indicating its potential to maintain oral microbial balance for prospective anticaries therapy.

## 5. Materials and Methods

### 5.1. Bacterial Strains and Chemicals

The tested bacterial strains, *S. mutans* UA159, *S. sanguinis* SK36, and *S. gordonii* DL1, were obtained from our laboratory. All strains were cultured in a brain–heart infusion broth (BHI; Oxoid, UK) at 37 °C (5% CO_2_). Celastrol was purchased from Beijing Solarbio company (Beijing, China) and was dissolved in dimethyl sulfoxide (DMSO) as 10 mg/mL stock solution and stored at −20 °C. The working solution of celastrol was freshly prepared before each experiment by 10× dilution of stock solution in BHI or DMEM. The final DMSO concentration in the culture did not exceed 0.02%. Catalase was purchased from Beijing Solarbio company (Beijing, China) and was dissolved in PBS as 1 mg/mL stock solution and stored at −20 °C before the evaluation.

For multispecies biofilm formation, bacterial suspensions were mixed to obtain an inoculum containing a defined microbial population consisting of UA159 (10^7^ CFU/mL), SK36 (10^7^ CFU/mL), and DL1 (10^7^ CFU/mL) in 1.6 mL of 1% sucrose-supplemented BHI medium in 24-well plates. The bacterial concentrations of inoculum were based on a previous study [41]. For monospecies biofilm formation, bacteria were inoculated at a concentration of 10^7^ CFU/mL in 1.6 mL of 1% sucrose-supplemented BHI medium in 24-well plates. The bacteria culture medium was refreshed every 24 h.

### 5.2. Minimal Inhibitory Concentration (MIC) and Minimum Bactericidal Concentration (MBC) Test of Celastrol

MIC and MBC of celastrol against UA159, SK36, and DL1 were tested by serial microdilution assays [62,63]. The starting concentration of microorganism was adjusted to 1 × 10^6^ Colony-Forming Units (CFUs)/mL. MIC was defined as the lowest concentration that inhibits bacterial growth completely, as compared to cells in control groups. Then, wells with no visible bacterial growth were sub-cultured for overnight incubation. Colony-Forming Unit (CFU) counts were conducted to identify the MBC. The MBC of celastrol was defined as the lowest concentration, which is capable in killing ≥ 99.9% of the initial bacterial inoculum.

### 5.3. Cell Cytotoxicity Test

Human oral keratinocyte (HOK, JENNIO, Guangzhou, China) cells were seeded in 96-well plates with a density of 1 × 10^4^ cells/well and cultured in DMEM for 24 h in 5% CO_2_ at 37 °C. The culture media were replaced with fresh media supplemented with celastrol. After incubation for 24 h, the culture medium was removed, and 110 μL of fresh medium containing 10 μL Cell Counting Kit-8 solution (CCK-8, Dojindo, Kumamoto, Japan) was added to each well and incubated for an additional 2 h, and then the absorbance at 450 nm was measured using SpectraMax^®^ iD5 (Molecular Devices, San Jose, CA, USA). Relative cell viability was calculated as a percentage of untreated controls [63,64].

### 5.4. Measurement of Bacterial Growth Curves

Bacteria strains were cultured overnight in BHI medium. Stationary phase cultures were diluted to a concentration of 10^7^ CFU/mL in fresh medium. Then, cultures were added to a 96-well plate. The plate was sealed with mineral oil, and bacteria were shaken at 100 rpm at 37 °C. The OD values of the bacteria culture were measured at 1 h intervals at 600 nm using SpectraMax^®^ iD5 reader.

### 5.5. Colony-Forming Unit (CFU) Counts

The number of viable bacteria in biofilm was evaluated by CFU counting. Biofilms were washed twice with phosphate buffered saline (PBS). Then, biofilms were harvested by vortexing in PBS buffer before conducting the serial dilution and spreading on BHI agar plates. The CFU counts were determined by averaging the number of counted colonies from three plates.

### 5.6. Lactic Acid and Water-Insoluble Glucan Measurement

For lactic acid measurement, biofilms were rinsed with cysteine peptone water (Sigma-Aldrich, Saint Louis, MO, USA) and incubated with 0.2% sucrose-supplemented buffered peptone water (Solarbio, Beijing, China). After incubation for 3 h at 5% CO_2_, at 37 °C, the lactic acid production was monitored with SpectraMax^®^ iD5 reader at OD_340nm_. Prior to calculating lactic acid production in each group, a standard curve was prepared based on a lactic acid standard (Supelco Analytical; Bellefonte, PA, USA).

Water-insoluble glucan was determined according to previous studies [23,65]. Biofilms were washed twice and harvested by scraping and vortexing. In order to remove water-soluble glucan, biofilm suspensions were centrifuged at 4 °C (8000× *g*, 5 min) and washed with PBS. The precipitates were resuspended in 1 mol/L NaOH and incubated for 2 h. The supernatant from the suspensions was incubated with 0.1% anthrone (dissolved in 80% sulfuric acid) for an additional 5 min at 95 °C. The water-insoluble glucan production was monitored at OD_625nm_.

### 5.7. SEM Observations of Biofilm Treated with Celastrol

Biofilm was formed on circular microscope coverslips. After incubation, biofilms were washed twice with PBS. Biofilm samples were fixed with 2.5% glutaraldehyde overnight at 4 °C for immersion fixation. Then, alcohol dehydration was conducted. The dehydrated samples were dried to a critical point, gold-sprayed, and observed using SEM (FEI, Eindhoven, The Netherlands).

### 5.8. DNA Isolation and Real-Time Polymerase Chain Reaction

Total DNA of biofilm was extracted using a TIANamp Bacteria DNA kit (TIANGEN, Beijing, China) according to the manufacturer’s instructions. The bacterial composition was further quantified by species-specific real-time quantitative polymerase chain reaction (qPCR) by the method described by previous studies [41,42,66,67]. TaqMan real-time polymerase chain reaction using a Premix Ex Taq (Probe qPCR) was used to quantify the absolute number of UA159, SK36, and DL1. The specificity of probes was confirmed by conventional PCR, and the standard curves of these bacteria were plotted for each primer/probe set by using threshold cycle values obtained by amplifying successive 10-fold dilutions of known concentrations of DNA, which stands for corresponding concentration of bacteria from 10^9^ CFUs to 10^4^ CFUs. The numbers of 3 strains were calculated based on standard curves generated using respective standard strains. The primers and probes are listed in Table 1.

### 5.9. Competition Assays on Agar Plate

To assess the inhibitory ability of SK36 and DL1 against UA159, a competition assays protocol described previously was used with modifications [7,68]. Briefly, 2 μL of overnight culture of SK36 or DL1 strains in BHI medium was inoculated onto a BHI agar plate as the pioneer colonizer. After aerobic incubation for 12 h, 2 μL of UA159 was inoculated next to the pioneer colonizer such that the colonies almost touched each other. The BHI agar plates were aerobically incubated for 12 h. The presence of a proximal zone of inhibition at the intersection with the pioneer colony represents growth inhibition.

### 5.10. H_2_O_2_ Measurements

The concentration of H_2_O_2_ generated in the biofilm supernatant was determined using a Hydrogen Peroxide Assay Kit (Beyotime, Shanghai, China) following the manufacturer’s guidelines. After celastrol treatment, 50 μL of biofilm supernatant sample was placed into a transparent 96-well plate, and then 100 μL of the detection reagent was added and incubated at 25 °C for 30 min. The absorbance at 560 nm was measured for determination of H_2_O_2_ production.

### 5.11. Real-Time Quantitative PCR (qPCR) of spxB

We performed qPCR to evaluate *spxB* gene expression in both SK36 and DL1. Bacterial culture at logarithmic phase was diluted in fresh BHI medium to approximately 10^8^ CFU/mL and treated with celastrol aerobically for 30 min. Then, samples were centrifugated at 4 °C, and supernatants were removed. Bacterial RNA was extracted using the MasterPure Complete DNA and RNA Purification Kit (Lucigen, Middleton, WI, USA). cDNA was synthesized from extracted RNA using a reverse transcription kit (Takara, Shiga, Japan). Specific primers used in this study are listed in Table 2. The expression of *spxB* was normalized to 16S rDNA gene transcription and was calculated using the 2^−ΔΔCt^ method.

### 5.12. Pyruvate Measurements

The concentration of pyruvate in the biofilm supernatant was determined using a Pyruvate Content Assay Kit (Solarbio, Beijing, China) following the manufacturer’s guidelines. After celastrol treatment, 75 μL of biofilm supernatant sample was placed into a transparent 96-well plate, and then 25 μL of the detection reagent was added and incubated at 25 °C for 2 min. The absorbance at 520 nm was measured using a spectrophotometer.

### 5.13. spxB Knockout Mutant Construction

In this study, we constructed *spxB* knockout mutant strain of SK36 and DL1 with IFDC2 cassette through allelic homologous recombination [69,70]. Briefly, an upstream 1kb fragment and a downstream 1kb fragment of *spxB* were amplified by PCR. Subsequently, the upstream and downstream fragments were combined in an overlapping PCR with erythromycin (erm) resistance cassette, and the PCR amplicons were transformed into DL1 and SK36, generating DL1 Δ*spxB* and SK36 Δ*spxB*, respectively. In detail, the primer pairs *spxB*-upF and *spxB*-upR were used for the upstream fragments. The downstream fragments were amplified using the primer pairs *spxB*-dnF and *spxB*-dnR. The erm resistance cassette was amplified by using the primer pair ermF and ermR. For overlap PCR, 50 ng of each purified PCR product was added directly to 22 μL of Primestar DNA polymerase reaction mixture (Takara, Shiga, Japan). The resulting PCR amplicon was transformed into SK36 and DL1 strains’ competent cells induced by their respective competence-stimulating peptides (CSPs). The deletion of *spxB* was verified by PCR with primer pairs checkF and checkR and sequencing. All primers are listed in Table 3.

### 5.14. Statistical Analysis

One-way analysis of variance (ANOVA) was used to investigate the significant effects of the variables, followed by Student’s *t*-test. Tukey post-test was applied to compare the data in each group. The *t*-test was performed for two groups. Data were considered significantly different if the *p* value was <0.05.

## Figures and Tables

**Figure 1 antibiotics-12-01245-f001:**
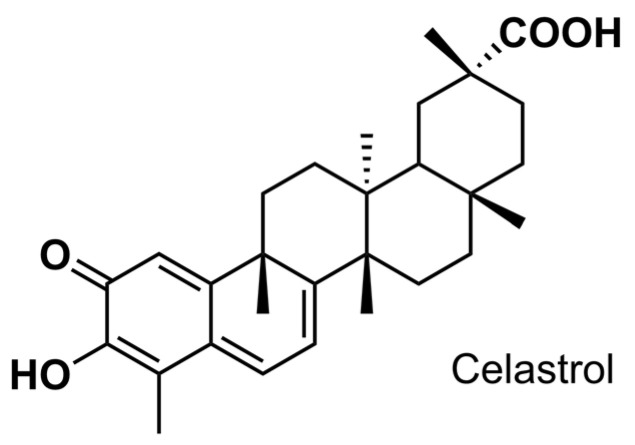
Chemical structure of celastrol.

**Figure 2 antibiotics-12-01245-f002:**
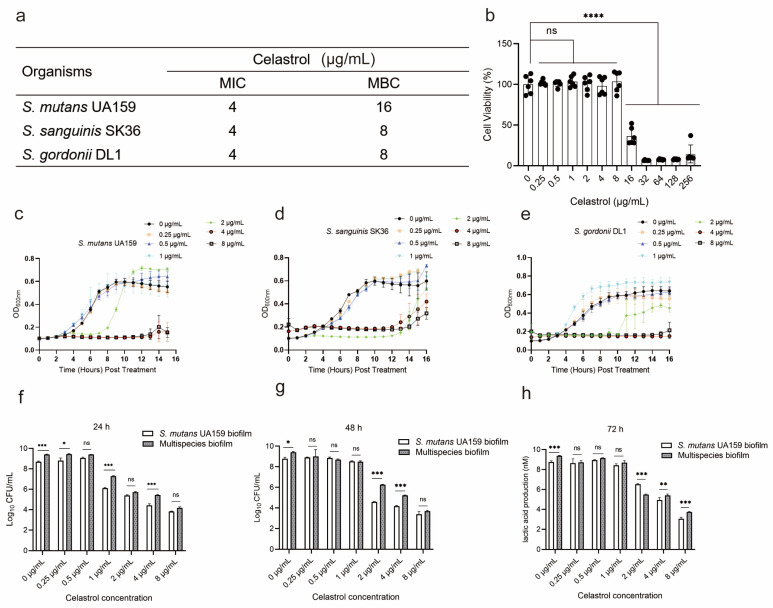
Biosafety and antibacterial properties of celastrol. (**a**) Minimum inhibitory concentration (MIC) and minimum bactericidal concentration (MBC) of celastrol against *S. mutans* UA159, *S. sanguinis* SK36, and *S. gordonii* DL1. (**b**) Cytotoxicity of celastrol against human oral keratinocyte (HOK) cells monitored by viability test. Cells were cultured with celastrol for 24 h (*n* = 5, mean values ± s.d.). Growth curves of UA159 (**c**), SK36 (**d**), and DL1 (**e**) treated with different concentrations of celastrol. Each value indicated the average of three independent determinations. CFU counts of the UA159 monospecies biofilm and multispecies biofilms cultured for 24 h (**f**), 48 h (**g**), and 72 h (**h**) with celastrol treatment. Each value indicated the average of three independent determinations. * *p* < 0.05, ** *p* < 0.01, *** *p* < 0.001, and **** *p* < 0.0001; ns, not significant.

**Figure 3 antibiotics-12-01245-f003:**
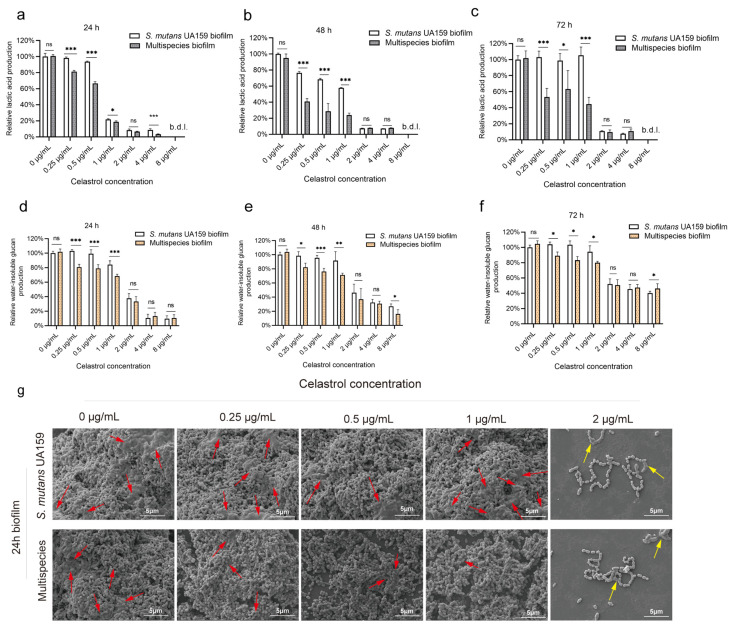
Celastrol, even at relatively low concentrations, could inhibit lactic acid production and water-insoluble glucan formation in multispecies biofilm. Relative lactic acid production of the UA159 monospecies biofilm and multispecies biofilms cultured for 24 h (**a**), 48 h (**b**), and 72 h (**c**) with celastrol treatment. Each value indicated the average of three independent determinations. Relative water-insoluble glucan production of the UA159 biofilm and multispecies biofilm cultured for 24 h (**d**), 48 h (**e**), and 72 h (**f**) with celastrol treatment. Each value indicated the average of three independent determinations. (**g**) SEM observations (5000× magnification) of UA159 biofilm and multispecies biofilm cultured for 24 h. Extracellular matrix and aggregated bacteria are indicated by red arrows. Bacterial lysate fragments are indicated by yellow arrows. b.d.l., below detection limit. * *p* < 0.05, ** *p* < 0.01, and *** *p* < 0.001; ns, not significant.

**Figure 4 antibiotics-12-01245-f004:**
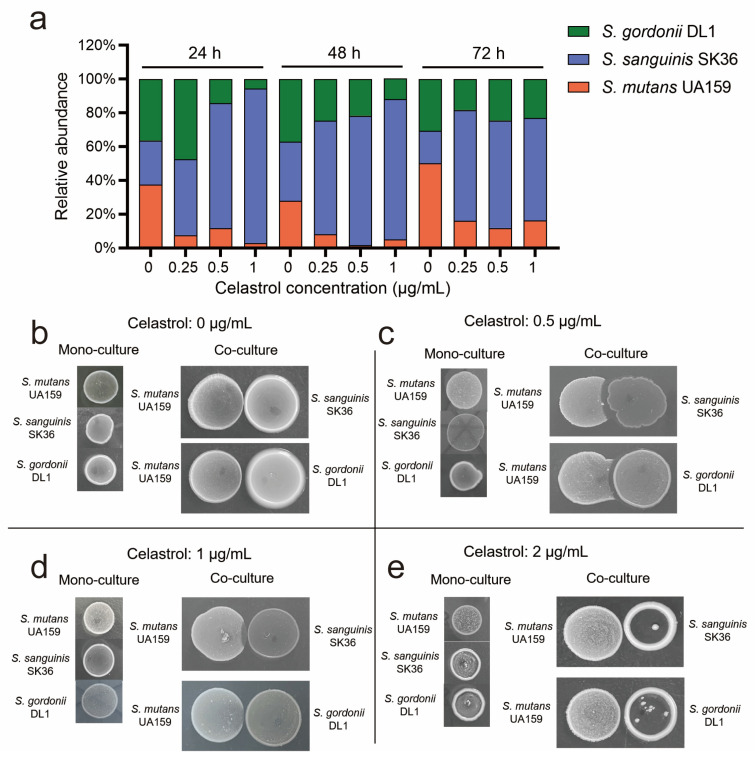
Celastrol reduced *S. mutans* proportion in multispecies biofilm by promoting streptococcal antagonism. (**a**) The ratio of UA159, SK36, and DL1 in multispecies biofilm, conducted by TaqMan real-time polymerase chain reaction, and each value indicated the average of three independent determinations. Growth inhibition of *S. mutans* UA159 by SK36 and DL1 in the presence of celastrol at 0 μg/mL (**b**), 0.5 μg/mL (**c**), 1 μg/mL (**d**), and at 2 μg/mL (**e**).

**Figure 5 antibiotics-12-01245-f005:**
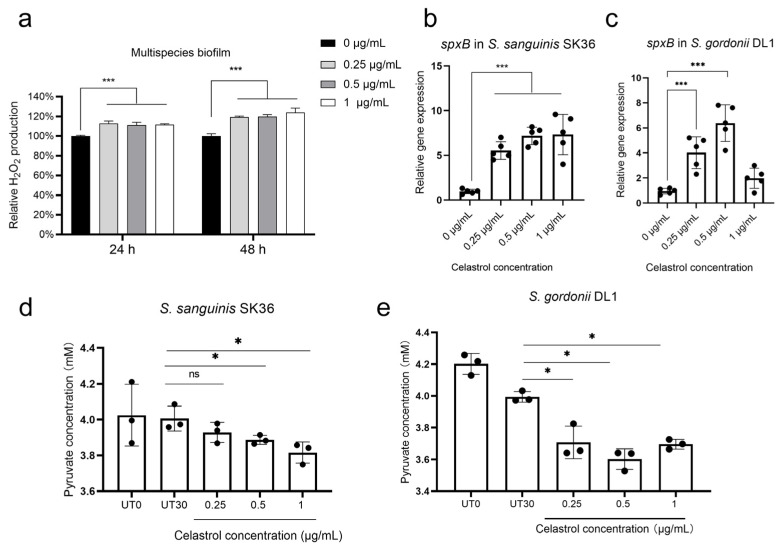
Low concentrations of celastrol activated SpxB in SK36 and DL1, leading to an increased level of H_2_O_2_ in multispecies biofilm. (**a**) Relative H_2_O_2_ production in multispecies biofilm treated with celastrol for 24 h and 48 h (*n* = 3, mean values ± s.d.). Relative expression of *spxB* in SK36 (**b**) and DL1 (**c**) treated with celastrol for 30 min (*n* = 5, mean values ± s.d.). Pyruvate concentrations were determined enzymatically in supernatants of planktonically grown SK36 (**d**) and DL1 (**e**) (*n* = 3, mean values ± s.d.). UT0 represents pyruvate concentrations at the time of celastrol addition; UT30 represents 30 min of growth in the absence of celastrol, and 0.25, 0.5, and 1 represent 30 min of growth at the corresponding concentrations of celastrol. * *p* < 0.05 and *** *p* < 0.001; ns, not significant.

**Figure 6 antibiotics-12-01245-f006:**
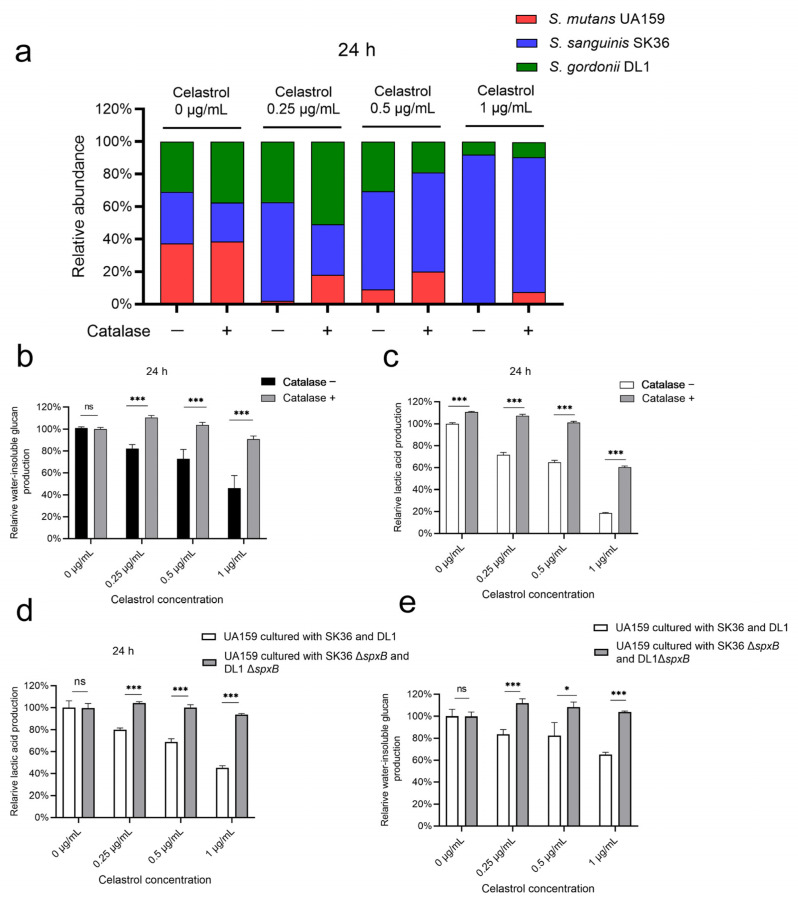
Celastrol thwarted *S. mutans* outgrowth by activation of SpxB and H_2_O_2_-dependent streptococcal antagonism. (**a**) The ratio of UA159, SK36, and DL1 in multispecies biofilm in the presence or absence of catalase, and each value indicated the average of three independent determinations. Relative lactic acid (**b**) and water-insoluble glucan (**c**) production of multispecies biofilms cultured for 24 h in the presence or absence of catalase (*n* = 4, mean values ± s.d.). When *S. mutans* UA159 cultured with SK36 Δ*spxB* and DL1 Δ*spxB* mutants, the relative lactic acid (**d**) and water-insoluble glucan (**e**) production of 24 h multispecies biofilms (*n* = 4, mean values ± s.d.). * *p* < 0.05 and *** *p* < 0.001; ns, not significant.

**Table 1 antibiotics-12-01245-t001:** Primers and probes used for species-specific real-time quantitative polymerase chain reaction.

Primers/Probes	Primer Sequence (5′–3′)	References
Primers:		
UA159-F	GCCTACAGCTCAGAGATGCTATTCT	Yoshida et al., 2003 [66]
UA159-R	GCCATACACCACTCATGAATTGA	
SK36-F	GAGCGGATGGCCAATTATATCT	Zhang et al., 2015 [41]
SK36-R	CCGGATGATGTCGGCAATA	
DL1-F	GGTGTTGTTTGACCCGTTCAG	Suzuki et al., 2004 [67]
DL1-R	AGTCCATCCCACGAGCACAG	
Probes:		
UA159	FAM-TGGAAATGACGGTCGCCGTTATGAA-TAMRA	Yoshida et al., 2003 [66]
SK36	FAM-TGTTCGGGCTCATGATA-Eclipse	Zhang et al., 2015 [41]
DL1	FAM-AACCTTGACCCGCTCATTACCAGCTAGTATG- TAMRA	Suzuki et al., 2004 [67]

**Table 2 antibiotics-12-01245-t002:** Primers used for real-time quantitative polymerase chain reaction.

Genes	Primer Sequence (Forward and Reverse)
*spxB* SK36	F: AATTCGGCGGCTCAATCG
	R: AAGGATAGCAAGGAATGGAGTG
*spxB* DL1	F: TTGCAGTAGGTTCAGGTGGT
	R: GGCAAGCTTCGTCAATCACT
16S rDNA SK36	F: AAGCAACGCGAAGAACCTTA
	R: GTCTCGCTAGAGTGCCCAAC
16S rDNA DL1	F: GCTTGCTACACCATAGACTG
	R: TCCTCATCACCATCCATAAAG

**Table 3 antibiotics-12-01245-t003:** Primers used for *spxB* knockout mutant construction.

Primers	Primer Sequence
SK36:	
*spxB*-upF	ATACTTGAGCAATACTTAG
*spxB*-upR	GAGTGTTATTGTTGCTCGGCATAATAACTCTCCTTCAATA
*spxB*-dnF	GGTATACTACTGACAGCTTCGGATTGCAATCACGCGCAA
*spxB*-dnR	CCATCTTCAGTATCATAC
checkF	CAGAAGCCGGTGTTTTAC
checkR	ATATTCACTTTCCGTTGT
DL1:	
*spxB*-upF	ATAATGAACGGGTGGCCCAAG
*spxB*-upR	GAGTGTTATTGTTGCTCGGAAGAATAACTCTCCTTCAA
*spxB*-dnF	GGTATACTACTGACAGCTTCTTCTTCTCGTCGAAAATCAA
*spxB*-dnR	CGCTACCATCTTCTGTGTCG
checkF	GGCCAGCTCAAAAGAAGC
checkR	GTTGAAATACACGCTACCATC
ermF	CCGAGCAACAATAACACTC
ermR	GAAGCTGTCAGTAGTATACC

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
