# Peer review of "Anticariogenic Activity of Celastrol and Its Enhancement of Streptococcal Antagonism in Multispecies Biofilm"

_antibiotics, 2023, doi:10.3390/antibiotics12081245_

Round 1

Reviewer 1 Report

Since antibiotic treatment of biofilm-associated infections is very often ineffective, a search for new therapeutic solutions is necessary. In this context, the research described in Antibacterial activity of celastrol…” is valuable from both scientific and practical point of view. However, I have some comments, which need to be addressed before the acceptance of the manuscript to be published in Antibiotics.

Major comments:

1.      Introduction (line 51) and whole text: The abbreviation EPS in context of biofilms means extracellular polymeric substances, including polysaccharides, proteins, lipids or extracellular DNA – Clarify or change the abbreviation used.

2.      Introduction (line 93-101): Placing of research results obtained in the Introduction is completely unnecessary, not to say incorrect – Delete or prepare additional section: Conclusions

3.      Results (Fig. 2 f-h): The lack of the differences in number of bacteria between 24 h-old and 72h-old control biofilm (untreated with celastrol) is very strange. It looks like streptococci did not multiply during the culture. How to explained?

4.      Materials and methods: Since celastrol was dissolved in DMSO, what was the final concentration of this toxic solvent? Complete this important information.

Minor comments:

1.      The first sentence of Introduction (line 29) and the third one (line 31) mean exactly the same – Delete one of them.

2.      Unify spelling: multi-species (text) or multispecies (figures)

3.      Materials and methods (sect. 4.2): Improve incorrect expansion of the abbreviation MIC

4.      Results (line 243): ” H2O2 production, The expression profile” – lowercase in “the”

5.      Discussion (line 311): “Here in this study” – Choose one: “here” or “in this study”

6. In reference no. 25 a number is repeated

Reviewer 2 Report

please attachment file. 

Doesn't exist.

Round 2

Reviewer 2 Report

I hope you have a good result.